# FaceUI: Leveraging Front-Facing Camera Input to Access Mid-Air Spatial Interfaces on Smartphones

Thuan Vo*
University of British Columbia - Okanagan, Canada

Marium-E- Jannat†
University of British Columbia - Okanagan, Canada

David Ahlström‡
University of Klagenfurt, Austria

Khalad Hasan§
University of British Columbia - Okanagan, Canada

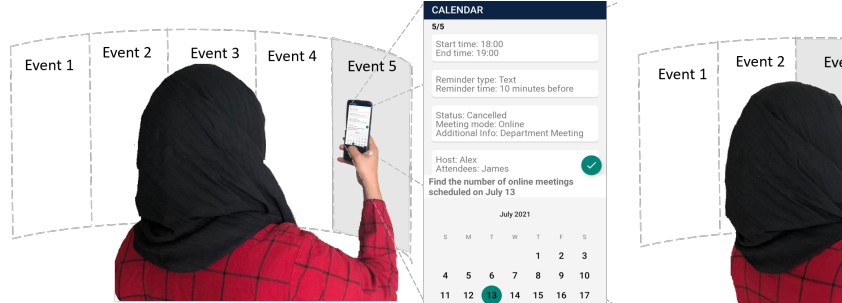 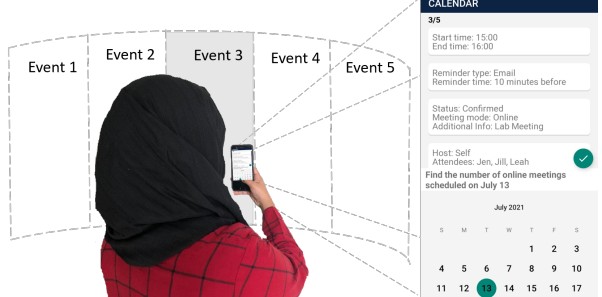

Figure 1: A FaceUI-based calendar app where the user accesses calendar events by moving the phone in mid-air around her face.

## ABSTRACT

We present FaceUI, a novel strategy to access mid-air face-centered spatial interfaces with off-the-shelf smartphones. FaceUI uses the smartphone's front-facing camera to track the phone's mid-air position relative to the user's face. This self-contained tracking mechanism opens up new opportunities to enable mid-air interactions on off-the-shelf smartphones. We demonstrate one possibility that leverages the empty mid-air space in front of the user to accommodate virtual windows which the user can browse by moving the phone in the space in front of their face. We inform our implementation of FaceUI by first studying essential design factors, such as the comfortable face-to-phone distance range and appropriate viewing angles for browsing mid-air windows and visually accessing their content. After that, we compare users' performance with FaceUI to their performance when using a touch-based interface in an analytic task that requires browsing multiple windows. We find that FaceUI offers better performance than the traditional touch-based interface. We conclude with recommendations for the design and use of face-centered mid-air interfaces on smartphones.

**Index Terms:** Human-centered computing—Visualization—Visualization techniques—Treemaps; Human-centered computing—Visualization—Visualization design and evaluation methods

## 1 INTRODUCTION

When using touch-based input on a smartphone, people typically hold the phone more or less in front of the face. This posture allows for easy visual access to screen content. Clearly, while interacting, having the phone in a stationary position seems preferable, and, accordingly, most smartphone interfaces assume a stable in-front-of-the-face posture. However, many people are also very skilled in

* e-mail: thuan.vo@ubc.ca

† e-mail: marium.jannat@ubc.ca

‡ e-mail: david.ahlstroem@aau.at

§ e-mail: khalad.hasan@ubc.ca

sub-optimal situations when the phone is not still in front of their face, such as to text while walking. In this paper, we explore how to design smartphone interfaces that require that the user deliberately moves the phone in the space in front of the face as part of the interaction. We use the high-resolution front-facing camera on a standard smartphone together with machine learning algorithms [10] to track the spatial location of the phone relative to the user's face. This allows us to integrate the large empty space in front of the user into new spatial interactions and user interfaces.

Prior research have explored ways to extend a smartphone's input capabilities by shifting the interaction space into the empty in-air space surrounding users' bodies or their mobile devices. For instance, Virtual Shelves [29] allows users to point their hand inside a hemisphere in front of their body to access a set of discrete virtual and invisible items, relying heavily on the users' spatial recall. Similarly, the Imaginary Interface [11] is a mid-air interface in front of the user's body that can be used for pointing and drawing activities. In more recent work, Hasan et al. [20] present the AirPane system and demonstrate how the mid-air space surrounding a mobile device can be used for browsing information in an e-commerce application. These and most other prior projects that demonstrate approaches to leverage around-body or around-device interactions rely on external tracking systems, which is not practical in real-life usage situations. Furthermore, most earlier projects are also limited in that they either do not provide any visual representation of the in-air space (and its interaction objects) at all, or they provide very limited visual information that typically is decoupled from the actual location within the in-air space.

We present FaceUI, an approach that avoids these shortcomings. FaceUI is a novel strategy that leverages mid-air space in front of the user. FaceUI uses a smartphone's built-in front-facing camera to detect and track the phone's position relative to the user's face. This self-contained tracking approach allows visual access to the in-air space since the screen content is updated according to the phone's in-air location and the virtual content at that location. Figure 1 shows a user who navigates a FaceUI-based calendar application.

To our best knowledge, ways to leverage face-centered in-air spaces to access virtual user interfaces (UIs) with off-the-shelf smartphones have never been explored before. With two user studies, we first investigate how the in-air space can be structured to accommo-

date virtual UIs used for information exploration on smartphones. We identify the comfortable phone-to-face distance range for accessing virtual UIs in the in-air space and suitable viewing angles for browsing and inspecting content that reside in the in-air space. We use this knowledge to design FaceUI-based calendar application. In a third user study, we evaluate users' performance in a calendar browsing task comparing our FaceUI-based calendar with a touch-based calendar interface. Our results show that the FaceUI-approach can offer considerable advantages compared to traditional touch-based interfaces. We end our exploration with showcasing further FaceUI-based applications.

Accordingly, our contributions include: 1) FaceUI, a novel face-centered spatial in-air interface-approach for off-the-shelf smartphones; 2) an exploration of suitable design parameters for FaceUI-based applications; 3) a performance comparison between a FaceUI application and standard touch interface in an analytic task; and 4) showcasing further promising FaceUI-enabled interactive applications that demonstrate the potential of face-centered smartphone interfaces.

## 2 Background and Related Work

We review prior work that has explored ways to design spatial interfaces, interaction spaces, and interaction techniques. These earlier projects inspired the design of our face-centered spatial user interface, FaceUI. The previous research closely aligned to components of FaceUI falls mainly under around-device interaction, on- and around-body interaction, and Face-Centered Input.

### 2.1 Around-Device Interaction

There has been substantial prior research work exploring the use of mid-air space around mobile devices. Researchers have demonstrated that the mid-air space can be used for novel interactions, such as for virtual content browsing and selection [14, 17, 18, 20, 25, 39], map navigation [19, 23], mode switching [23] and typing [32]. For instance, AD-Binning [17] leveraged the empty 2D space around a smartphone to off-load and browse content into the space. They further showed that the mid-air space could facilitate faster access to items than the standard touch input. In a similar work, Hasan et al. [20] showed that 3D in-air space around a device could be used for browsing m-commerce applications. Researchers also investigated ways to track users activities around the device with commercial tracking solutions (e.g., Vicon tracking [18–20]) or using different cameras or sensor-based solutions (e.g., depth camera [7, 26] , distance sensor [6, 25]). Though these solutions offer precise motion capture data, they require either environments, users or devices to be instrumented with sensors. This makes mobile devices less portable to be used in public spaces.

### 2.2 On- and Around-Body Interaction

Prior work investigated ways to use the on- and around-body space for designing novel interaction with devices [3, 5, 9, 29]. For instance, researchers [11–13] explored the use on body locations such as palm to access on-screen contents. Imaginary Phone [12] used user's palm as the input surface for iPhone. In a similar work, Gustafson et al. [13] investigated palm-based imaginary interface for supporting visually impaired users. Imaginary Interfaces [11] allowed users to perform spatial interaction on empty palm and without visual feedback. In addition, palm has been used for trigger pre-defined functions [28], to perform 3D rotation [26], or use it as an input space for augmenting keyboards [36]. Similarly, researchers explored the skin as an interactive touch surface [16, 41, 42]. They commonly used external depth cameras to detect and track hand and finger activities such as tapping and sliding on body parts.

Researchers also investigated using the mid-air space around the body as a novel interaction space. For instance, Virtual Shelves [29] demonstrated that the mid-air space in front of users could be used

to trigger shortcuts. With a study, they showed that users could recall shortcuts by moving their phone into a $7 \times 4$ grid on a circular hemisphere in front of them. Yee et al. [40] designed a solution allowing users to move the mobile phone to different locations around the body and change the on-screen content based on the device's location relative to the body. Ens et al. [9] designed Personal Cockpit leveraging the around-body space to display virtual windows in an head-worn displays. In a similar work, Babic et al. [5] explored Gesture Drawer, an one-handed interaction technique allowing users to define and interact with self-define imaginary interfaces while moving their hand to interact with the interfaces. Researchers have also investigated mid-air spatial interface specific to applications in mixed-reality [9, 31, 37], for games [35], workspace navigation [24]. For instance, Lubos et al. [31] introduced kinespheres, an mixed-reality based body-centric spatial interface within arm's reach. They received positive feedback from users on using their method compared to traditional head-centered interaction for mixed-reality. Yan et al. [37] explored an eyes-free target acquisition technique for mixed-reality by placing the targets in around-body space. Way Out [35] is a game scenario where players can navigate through an omni-directional panorama scene by moving the device around the body using the built-in motion sensors in smartphones. In a recent work, Kim et al. [24] demonstrated image and map zoom-in and zoom-out using the vision-based interface OddEyeCam, that detects and tracks the location of the mobile phone with respect to the user's body using external sensors such as wide-view RGB cameras and narrow-view depth cameras.

### 2.3 Face-Centered Input

Prior research investigated using head and face movements as an input to design new face-centered interactions on devices. For instance, Zhao et al. [43] used a combination of facial movements, device motion and touch for designing face-centered interaction techniques on smartphones. Kumar et al. [27] leveraged eye gaze to scroll mobile phone contents. Yang et al. [38] used a face interpretation engine for enabling face-aware applications for smartphones using the phone's front-facing camera and built-in motion sensors. Similarly, Babie et al. [4] designed Simo that used head movement as input for pointing on a distant large display. Instead of using external cameras, they used the smartphone's front-facing camera to detect face orientation. Rustagi et al. [33] explored touchless typing using head gestures detected by the smartphone's front-facing camera and used them to type on an on-screen QWERTY keyboard. We also observed that the smartphone's front-facing camera could be used to design new strategies such as for rotation of on-screen content on mobile devices by detecting direction changes of objects from the camera view [1, 2, 8].

The manifold opportunities of spatial mid-air interfaces, as demonstrated by earlier projects, inspire us to continue on this promising path. However, in opposite to most previous projects which use external sensors or cameras to identify interaction gestures, we are interested in using a self-contained tracking mechanism to detect in-air movements. Similar to a few face-tracking systems [4, 33, 43], our FaceUI-approach also uses the front-camera of a smartphone to detect changes in the relative positions of the user's face and the phone. However, FaceUI differ from earlier systems in that it does not rely on any other sensors than the smartphone's front-facing camera. Furthermore, we aim at interactions where the user keeps the head still while moving the phone. Earlier approaches [4, 33, 43] require the user to do the opposite, to move the head while holding the phone in a fixed position. In this way, we intend to create the sensation of a hemispherical interaction space that is anchored in front of the user's face but moves along with the user (through the self-contained tracking). In this first exploration of such an interaction hemisphere, we focus on using virtual application windows that are located inside the hemisphere. When the user

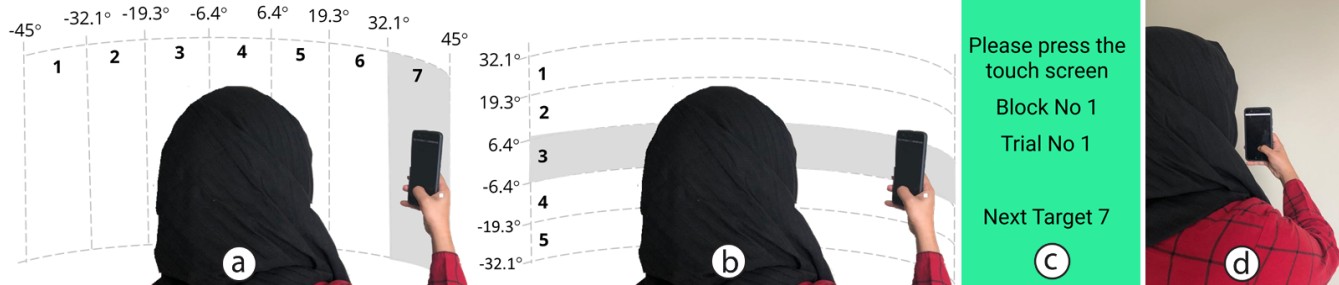

Figure 2: Study 1 task. Using (a) horizontal and (b) vertical mid-air movements to select invisible in-air items. (c) Task prompt. (d) A participant holding the phone in the neutral start position, straight in front of the face.

moves the smartphone to a location inside the hemisphere, the content of the virtual window that resides at that location is displayed on the smartphone's screen. When the user re-positions the smartphone inside the hemisphere the screen displays the content of the virtual window that resides at the new position.

Next we describe a few central aspects of the face-detection software and the setup we used in our user studies. After that we present our three studies in turn and order.

## 3 FACE DETECTION SOFTWARE AND STUDY SETUP

The self-contained tracking software facility we developed for our FaceUI-approach is based on the Face Detection API [10] in Google's ML Kit (*Machine learning for mobile developers*). The API provides a comfortable and reliable way to track the position and orientation of a smartphone relative to the user's face when the front-facing camera is used. Among the available face-tracking related measures, our software relies on *yaw* data (the smartphone's movements to the left or to the right relative to the detected face), *pitch* data (the smartphone's up and down movements in the vertical direction relative to the detected face), and *distance* data (the current distance between the detected face and the camera lens of the front camera on the smartphone). Our software does not use any *roll*-related information. The Face Detection API delivers 0° for both yaw and pitch when the user holds the phone straight in front of the face.

Restrictions related to COVID-19 prevented us from meeting our study participants face-to-face. Instead, we conducted our studies remotely using teleconferencing software. Accordingly, our participants were required to have a laptop or a desktop computer with a stable Internet connection, a microphone, loudspeakers, and a webcam. In the studies our participants used their own smartphone to run the study software. The study software was designed for any phone running Android 4.2 to 11. Our participants received the study software (i.e., the apk file) and all necessary instructions over email and we guided them through the installation process in the beginning of the study session. The data logged during a study session was automatically transferred from the participant's phone to a Cloud-Firestore data base when the participant had completed the last study tasks.

We ran all of our three studies remotely, where participants used the study apps on their smartphones in the wild as opposed to the controlled lab environment. All participants sat in front of their webcam while completing the study tasks. In each study, a study session lasted approximately 45 minutes, including instructions, practice trials, timed study trials, breaks, and completion of questionnaires. As the study apps were designed for the Android platform, we only recruited participants who possessed an Android smartphone.

## 4 STUDY 1: EXPLORING DIRECTION AND DISTANCE

Prior research has reported arm fatigue and 'heavy arm'-issues related to mid-air interactions [15] and that working with a bent arm in mid-air is more comfortable and less strenuous than working with a stretched arm [21]. Since FaceUI involves mid-air hand movements arm fatigue is a potential problem. Moreover, with FaceUI, the mid-air movements need to be constrained such that the user's face is inside the front-camera's field of view.

With FaceUI, we envision the mid-air interaction space as a semi-circular space in front of the user's face. Through a pilot test (with five participants) we found that the face tracking works best when the phone is between 5 and 80 centimetres away from the user's face and the user moves the phone within a longitudinal range of 90° (from -45° to the left of the user's face to 45° to the right of the user's face) and a latitudinal range of 70° (from -35° below the user's nose to 35° above the user's nose). Whereas we know that movements inside this space are accurately tracked, we do not know how accurately, fast, and comfortably people can navigate around in this mid-air space. Accordingly, we want to chart out the suitable dimensions and the granularity of the mid-air interaction space for FaceUI in our first study.

### 4.1 Study Design and Study Task

We oriented the study task and study design of our first study according to previous projects that have explored the dimensions and the granularity of the mid-air space in front of the user, e.g., the Virtual Shelves [29] and AD-Binning [17] projects. We used a simple item selection task where a trial consists of moving the smartphone to a specified position in mid-air to select the virtual item at that position. We investigated horizontal movements and vertical movements when the phone is close or far from the user's face.

Figure 2 visualizes the study task and setup. We divided the mid-air space along the horizontal into seven equally wide one-dimensional regions – or items –, each 12.85° 'wide' (Figure 2a). We used five one-dimensional regions – or items – in the vertical direction, each 12.85° 'high' (Figure 2b). From a user's perspective, the size of these items in the air in front of the face depends on the distance between the phone and the face: the further away from the face, the larger the item becomes. Accordingly, we decided to also test movements (horizontal and vertical) when performed close to the face and far away from the face.

#### 4.1.1 Study design

With this, we arrive at two independent factors for our study: 1) movement *Direction*: horizontal and vertical, and 2) *Distance*: close and far. Close represents the distance range within which participants commonly and comfortably hold their phone when accessing on-screen content with touch. We regarded any distance beyond that range as far. However, where the comfortable range ends is likely

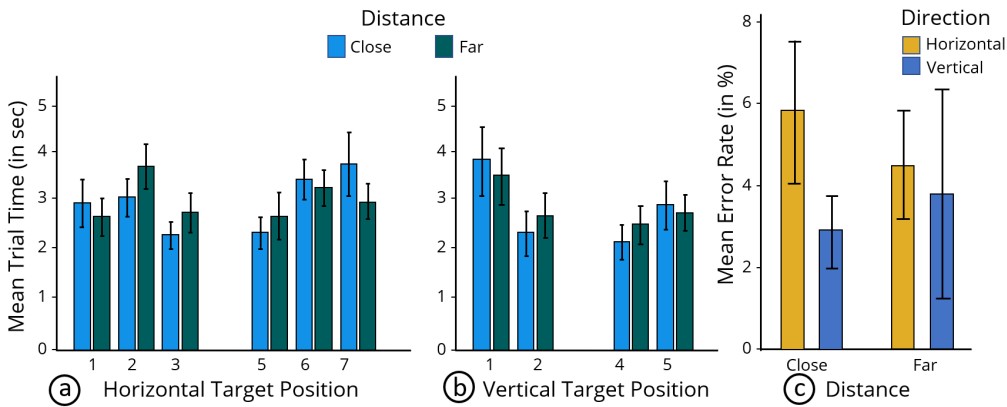

Figure 3: Result of Study 1. Mean trial time for close and far distance in (a) the horizontal direction and (b) in the vertical direction. (c) Mean error rate for the horizontal and vertical directions at close and far distances. Error bars: 95% CI.

to differ between participants (depending on arm length and preference). Therefore, it is critical to have a user-depended threshold value rather than using a common value for all participants. We calibrated the individual value for each participant in the beginning of the study session. We asked the participant to provide us the phone-to-face distance where it started to feel awkward and less comfortable when moving the phone in front of the face. Once the phone had reached these locations, the study app showed the distance between the face and the phone (in centimetres) on the screen. We asked participants to keep their head static and move the phone left, right, middle, up, and down and share the distance data. We calculated the participant's upper value for the 'close' (i.e., comfortable) distance for the horizontal direction by averaging the left, right and middle values. For the vertical direction we averaged the up, down, and middle values.

We used a within-subjects study design. All participants performed four series of six blocks of trials, one series of blocks with each of the four *Distance-Direction* combinations: close-horizontal, close-vertical, far-horizontal, and far-vertical. Blocks in the horizontal direction consisted of six trials, one trial for each of the six target items (1, 2, 3, 5, 6, and 7, cf. Figure 2a) in random order. Blocks in the vertical direction consisted of four trials, one trial for each of the four target items (1, 2, 4, and 5, cf. Figure 2b) in random order. Accordingly, each participant performed 120 trials: one block series of 36 horizontal trials at close distance + one block series of 24 vertical trials at close distance and one block series of 36 horizontal trials at far distance + one block series of 24 vertical trials at far distance. Half of the participants started with the two block series at close distance and then completed the two block series at far distance, the other half used the other order. The order of the two directions-series within a distance was random.

### 4.1.2 Task procedure

To start a trial, the participant moved the phone to the middle region, straight in front of the face. In the horizontal direction, this corresponded to Item 4 in Figure 2a. In the vertical direction Item 3 in Figure 2b was used as the start region. Once the participant moved the phone inside the start region the screen turned green and displayed information for the upcoming trial, including the target prompt with the item number to select next, as shown in Figure 2c. A trial started when the participant pressed down the thumb on the screen. If the phone was moved outside the start region before pressing down with the thumb, the screen turned red and showed instructions to move the phone into the start region. A thumb-press in the start region started timing for the trial. A selection was registered and the trial time stopped when the thumb was released after

having moved the phone into one of the items (or regions) outside the start region. Speech output informed if the participant selected the correct item or not by playing "Correct selection" resp. "Wrong selection". Erroneous trials were re-queued at a random position among the unfinished trials within the current block.

During a running trial we relied on audio to inform participants about the current position of the phone. The app provided speech output when i) the phone entered a new item, by saying the number of the item, when ii) the participant moved the phone at the wrong distance, by playing "Move the phone further away" in far conditions or "Move the phone closer" in close conditions, and when iii) the face tracking software lost track of the face, by playing "Face out of camera view". Working with audio guidance was important: in our first study, we wanted to focus on the motoric aspects and movement properties that determine the dimensions of FaceUI's interaction space. We wanted to exclude aspects that relate to how well a user can read screen content while moving the phone in mid-air space, such as the size of screen content and the viewing angle and distance. We return to such visual issues in our second study.

### 4.2 Participants

We recruited twelve right-handed participants (mean age 27.08 years, s.d. 5.98, 6 male) via on-campus flyers and word-of-mouth. All participants were daily smartphone users.

### 4.3 Results

We first report on results regarding participants' comfortable phone-to-face distance (calibrated in the beginning of a study session) that served as the basis for each participant's individual threshold value that separated the close distance from the far distance. After that we report on trial time, error rates, and subjective ratings.

#### 4.3.1 Close/far threshold value

Across all participants, the average face-to-phone distance where movements started to feel less comfortable was 39.31 cm (s.d. 7.61) for horizontal phone movements and 39.38 cm (s.d. 7.34) for vertical movements. This critical threshold varied a lot between participants. In the horizontal movement direction it was between 30 and 61 cm and in the vertical direction between 28 and 61 cm (only one participant had values greater than 50 cm).

#### 4.3.2 Trial time

The trial time analyses are based on error free trials only. Figure 3a shows the mean trial time for each target position at both distances in the horizontal movement direction and Figure 3b shows the corresponding results for the vertical direction. Mean trial times for close

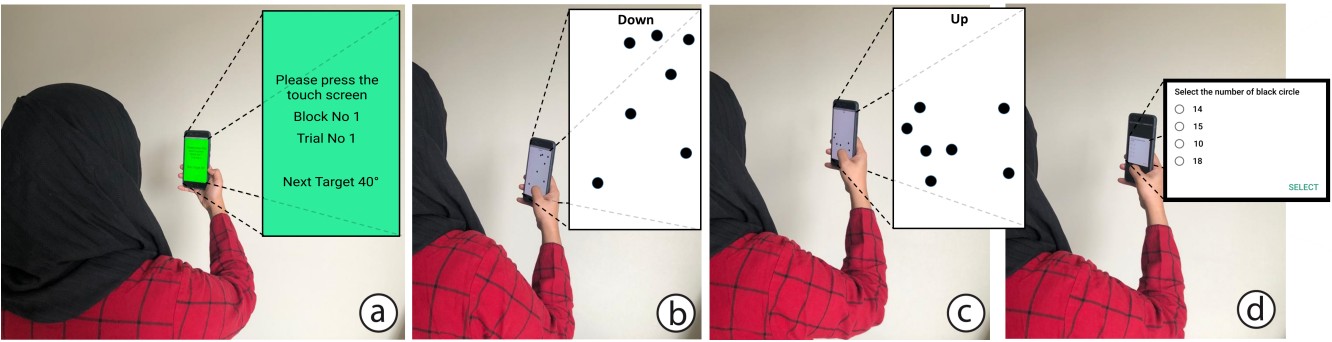

Figure 4: Study 2 setup. (a) The green screen showing the instruction of locating the phone to a 40° viewing angle on horizontal plane, (b) Participant located the phone to downward at 40° viewing angle on horizontal plane and, (c) located the phone to upward at 40° viewing angle on horizontal plane, (d) selecting the total number of black dots on down and up screen at 40° viewing angle on horizontal plane.

and far (across the two directions) were 2.93s and 2.99s, respectively. The overall mean trial time (across the two distances) for the horizontal and vertical directions were 3.01s and 2.90s, respectively. A $2\times2$ RM-ANOVA showed that there was no significant difference between the two distances ($F_{1,11} = 0.09$, $p = 0.76$) or between the two directions ($F_{1,11} = 0.29$, $p = 0.59$). A one-way RM-ANOVA (independent factor block) indicated that participants became faster during the course of the study with significantly longer trial times in the first block of trials than in last two blocks ($F_{5,55} = 5.85$, $p < 0.001$). The mean trial time decreased from 3.43s in Block 1 to 2.81s and 2.72s in Block 5 and 6, respectively.

In Figure 3 we also see a clear and expected pattern regarding the different target positions: given further phone-movement distances, selecting item at positions close to the start position (Position 4 for horizontal movements and Position 3 for vertical movements, cf. Figure 2) was quicker than selection items further away. This pattern we see for movements in both directions and at both the close and far distances.

### 4.3.3 Error rate

Figure 3c shows the mean error rates for the four *distance×direction* combinations. A Friedman test identified a significant difference among the combinations ($\chi^2(3, N = 12) = 8.95$, $p < 0.05$) and post-hoc Wilcoxon tests (Bonferroni adjusted $\alpha$-level from 0.05 to 0.008) revealed that the close-vertical combination was significantly less error prone than close-horizontal combination and that there were no other pairwise differences.

### 4.3.4 Subjective feedback

We asked participants to rate the two directions and the two distances according to their overall preference on a 5-point scale with $1 = bad$, $3 = neutral$, and $5 = good$. We found an unsurprising and strong preference for the close distance with mean rating 4.52 compared to the far distance with mean rating 1.91. Participants were not that decided in their opinions regarding the two movement directions. They rated the horizontal movement direction only slighter better than the vertical direction, mean rating 4.23 vs. mean rating 3.1.

### 4.4 Summary

Results from the subjective feedback indicate that participants had a slight preference for horizontal movements over vertical movements. However, our analyses also revealed that there is no significant difference between the movement directions in regard of trial time. But we see a clear, and unsurprising advantage for the close distance over the far distance. Accordingly, for our future FaceUI explorations, we learn that people are sensitive regarding the phone-to-face distance and that FaceUI-based applications should avoid requiring user to use large phone-to-face distances. Consequently, in our we continue

utilizing regions along both the horizontal and vertical directions. However, we observed increased trial time with items located in certain vertical regions, e.g., Item 1, than others. This warrants further investigation into factors such as visual angles that could influence users' performance when reading screen content when holding the phone in such regions.

## 5 STUDY 2: EXPLORING TARGET REGION AND ANGLE

Application interfaces that are placed in FaceUI move in both horizontal and vertical regions with the user's head movement along with the same regions. Therefore, to read content that is located to the right or left on the FaceUI, a user needs to keep their head static and move their eyes to read the content. Prior research [34] showed that such eye movement could cause eyes fatigue, pain and tiredness. Therefore, in this study, we explored suitable viewing angles where users can comfortably access on-screen items on smartphones.

### 5.1 Participants

We recruited fourteen right-handed participants (mean age 26.78 years, s.d. 6.07, 7 male) via on-campus flyers and word-of-mouth. All participants were daily smartphone users. None of the participants had participated in Study 1.

### 5.2 Factors

We considered two factors, target region and target angle, described as follows.

#### 5.2.1 Target region

In this study, we considered placing items in two regions – vertical (up and down) and horizontal (left and right). Similar to the first study, we kept the middle location reserved as the starting point of a trial.

#### 5.2.2 Target angle

We decided to place a set of targets at angles both in horizontal and vertical regions. With a pilot study, we choose to place items at $\pm20°$, $\pm30°$, $\pm40°$ and $\pm50°$ angles where positive and negative angles indicate items to the right and left regions, respectively. Results from our pilot study showed that participants were not able to see items that are located above +30° in the up. Additionally, any items placed below -40° angles for the downward region were not accessible as the phone gets very close to the body. Therefore, we used +20° and +30° for the up region, and -20°, -30° and -40° for the down region.

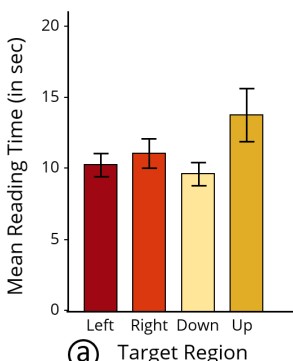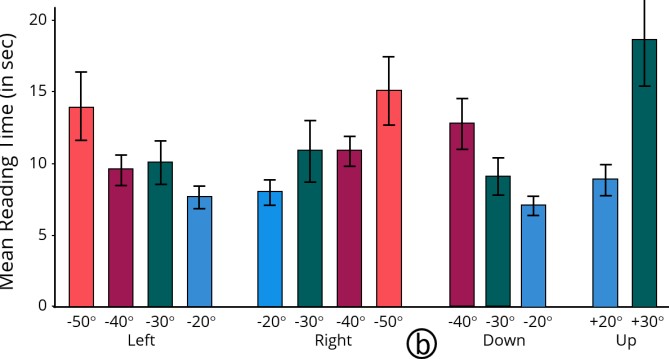

Figure 5: Result of Study 2. (a) Mean reading time for left, right, down and up target regions and (b) for target angles in each target region. Error bars: 95% CI.

## 5.3 Procedure and Tasks

At the beginning of a trial, the participant was required to move the phone to the middle position (straight in front of the face). As long as the phone was still and outside the middle position, the screen remained red and contained instructions to move the phone to the middle position. Once the phone was inside the middle position the screen turned green and displayed the target prompt for the next target (along with block and trial counts), as seen in Figure 4a. Participants were asked to press on the screen with their thumb and move their phone at the target angle while keeping the thumb on the screen. Tapping on the screen also started a timer. The study application then removed the on-screen instructions, replaced with an empty black window, and kept it until participants moved the phone to the target angle. We used circle counting tasks in this study, where participants were required to count the number of circles presented in two windows located at a target angle. For instance, if the target angle was +30° in the right region, we placed two more windows above and below the vertical plane defined by the user's eye. Participants could only see the windows once they reach to the instructed region and angle. We asked participants to keep the head still and move only the phone during the study. They could now move their phone up and down (for horizontal region) or left and right (for vertical region) to access the windows while keeping the phone in the target angle (Figure 4b-c). The windows contained a random number of non-overlapping black circles between 12 and 16. Participants were required to count the total number of circles seen in both windows. Once they believed having counted all circles in both windows, they were asked to lift off their thumb from the touchscreen. This action further poped up a window containing multiple options for the summation results (Figure 4d). Once they selected the correct answer, the application stopped the trial time, provided voice feedback on whether the answer was correct and displayed the instruction for the next trial on the screen. If incorrect, the app stopped the trial time, provided audio feedback, and re-queued the trial at a random position among unfinished trials within a block of trials. Participants were then required to move the phone back to the middle and continue trials until all the trials were finished. Note that for either case, the app sent trial-related information (e.g., task completion time, correctness) to a database server.

Each participant completed six blocks of trials with each of the *Target Region* (left, right, up and down) where one block contained one trial for each of the *Target Angles*. Therefore, each participant performed 78 error-free trials (24 trials for left, 24 for right, 12 for up and 18 for down region). The presentation order of the target region were selected randomly between participants and the angles were presented in a random order. Participants were provided with 2 blocks of practice trials. After completing all the trials, we collected the participants' feedback on their preferences on *Target Region* and *Target Angles*. This study required participants around 45 minutes to complete all the tasks.

## 5.4 Results

Instead of analyzing the trial time, we were interested in the time that participants spent on counting the circles rather than moving the phone to the target position and taking time to answer questions. We called this time as *Reading time*.

### 5.4.1 Reading time

We used repeated measures ANOVA and post-hoc pairwise comparisons to analyze reading time. Results showed that *Target Region* had significant effects on reading time ($F_{3,39} = 3.24$, $p < 0.05$). Figure 5a shows the mean reading time for all four regions: left (mean 10.40s) and right (mean 11.16s), down (mean 9.76s) and up (mean 13.84s). Post-hoc pairwise comparisons showed that Down was significantly faster than Up while accessing the items. No other pairwise difference was found.

We also analyzed the reading time for each target angle on each target region. Figure 5b shows the mean reading time for all target angle on each target region. Target angles in *Right* showed significant effects on reading time ($F_{3,39} = 6.71$, $p < 0.001$). Post-hoc pairwise comparisons between target angles showed that targets at 50° angle was significantly slower than target angles at 20°. There were no other pairwise statistically significant differences. We also observed that the target angles in *Left* target region had significant effects on reading time ($F_{3,39} = 7.03$, $p < 0.001$). Similar to the right region, targets at 50° angles were significantly slower than targets at 20°. There were no other statistically significant differences between the angles.

Like left and right regions, target angles in *Up* region showed significant effects on reading time ($F_{3,39} = 23.39$, $p < 0.001$). Targets at 30° angles were significantly slower than targets at angle 20°. Target angles in *Down* region also showed significant effects on trial time ($F_{2,26} = 21.12$, $p < 0.001$). Targets at 40° angle were significant slower than targets at 20° and 30°. We did not find any other statistically significant differences.

### 5.4.2 Subjective feedback

Participants rated each target region using a 5-point Likert scale. They preferred the right target region (mean rating 3.85) most, followed by down (mean rating 3.78) and left (mean rating 3.07). Up target region was rated as the least preferred region to access items (mean rating 2.28).

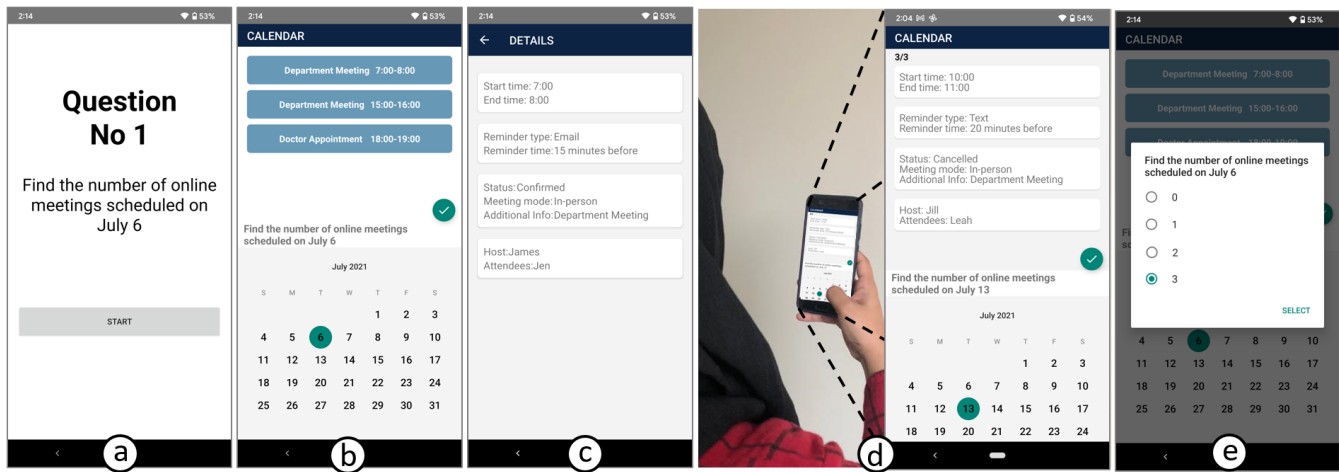

Figure 6: Calendar app interface for Study 3. (a) A trial starts with a query prompt, e.g., *"Find the number of online meetings scheduled on July 6"*. A tap on the start button opens a new window (b) with calendar dates at the bottom of the screen. With the touch interface, a tap on a date shows the events scheduled on that date at the top of the screen. The participant can further inspect an event by tapping on it. This opens a detail window (c) which displays the details for that event. The participant can return to the previous window by tapping the back button or by swiping left. (d) With FaceUI, the detail windows for the events on the selected date are placed as virtual windows in front of the user who can access these by moving the phone in mid-air. Both with FaceUI and with the touch interface, when the participant has inspected the event details and found the answer to the task query, the participant taps a check button to open a popup window (e) to select the answer from a list, which ends the trial.

## 5.5 Summary

The results shows that accessing the screen contents visually is slower when the phone is positioned in the upper region (up) than in any other region (the lower region, the left and the right regions) of the in-air space. For the target angles for each region, we see that participants' performance degrades significantly at the highest angle in each region. Accordingly, we suggest to avoid the extreme angles, 50° for both right and left, 30° for up, and 40° for down when designing a application that uses FaceUI.

## 6 STUDY 3: FACEUI PERFORMANCE IN AN ANALYTIC TASK

In our previous two studies we explored different design factors, such as target angle and target region, that could potentially influence users' performance with FaceUI. In this study, we evaluate a practical usage scenario with FaceUI where the user is required to browse multiple windows to retrieve information. Consequently, we designed a calendar app and compared users' performance when using its FaceUI version and when using a its touch-based version.

## 6.1 Participants

We recruited twelve right-handed participants (mean age 25.5, s.d. 5.23, 6 male) via on-campus flyers and word-of-mouth. All participants were daily smartphone users. None of the participants had participated in Study 1 or in Study 2.

## 6.2 Task, Procedure, and Design

The analytic task requires participants to review information in a calendar before reaching a decision. A trial starts with displaying a query prompt along with a start button, such as *"Find the number of online meetings scheduled on July 7"*, as seen in Figure 6a. After reading the question, the user taps on the start button, which starts the trial time and opens a new window. The new window contains calendar dates for one month at the bottom of the screen. Once the user taps on a date, the date is highlighted in green and a number of calendar events scheduled on that specific date is displayed at the top of the screen, as shown in Figure 6b. To determine a reasonable number of calendar events to add for each date, we surveyed students and faculty members and found that they commonly have 3

to 5 events (e.g., classes or meetings) per day, excluding weekends. Consequently, we used 3 to 5 events, each represented with an event title and time (e.g., *"Department Meeting 15:00-16:00"*), for each day except for Saturday and Sunday. Once the participant taps on an event, the app opens a detail view in a new window containing information about that event, as shown in Figure 6c. The design of detail view is inspired by Android's generic calendar application that contains event title, event time, names of people hosting or attending the event, the event type and mode (weekly meeting, online/in-person), reminder-related information (e.g., reminder type and time). After checking the detail view, the participant can either tap the "back" button or swipe left to return to the previous screen to view the other events on that day.

With FaceUI, a trial also starts with the screen displaying a query prompt and a start button. Once the user taps the button, trial time starts and a new window shows calendar dates at the bottom of the screen (Figure 6d). While designing FaceUI, we leveraged the empty mid-air space in front of the user to accommodate virtual windows for frequent browsing. Accordingly, we now use FaceUI in conjunction with touch input: touch is used to select a calendar date and FaceUI is used for browsing the detail views of the events taking place that date by moving the phone in mid-air space. As we have at most five events for a date (according to our small survey), we place the corresponding detail views between +40° and -40° in the horizontal direction in mid-air rather than placing them in a grid. We instructed the participant to keep the head still and to move the phone to browse the detail views in mid-air space. Once the participant selects a date (using touch), FaceUI shows the details of an event (i.e., its detailed view) on the top half of the screen. Now the participant can call in the details of the other events taking place that date by moving the phone horizontally in mid-air, as visualized in Figure 6d. Phone movements in the vertical direction are ignored.

When the participant believes having found the answer to the task question (after having inspected the events' detail views), the participant submits the answer by calling in an answer window (Figure 6e) and then selecting one of the listed answer alternatives. The submission step is the same for both the touch interface and for FaceUI: the participant taps a check button (the green button in the middle of the screen in Figure 6b and d) to see the answer window

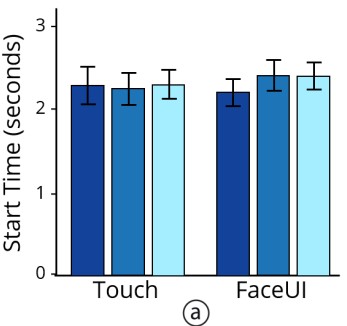
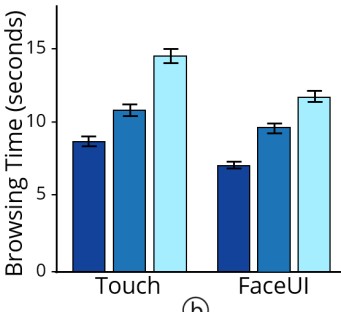
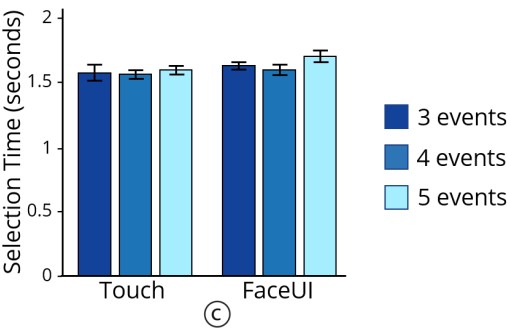

Figure 7: Result of Study 3. Mean (a) start time, (b) browsing time and (c) selection time. Error bars: 95% CI.

and selects the answer using finger taps. The app provides audio feedback on the correctness of the answer. If the answer is correct, the trial time stops, and the query prompt for the next is displayed. If incorrect, audio feedback for incorrect answer is provided, trial time stops, the trial is marked as erroneous, and the participant is required to redo the same trial, from the beginning (i.e., with the query prompt along with the start button).

The study used a 2×3 within-subjects design for factors *Interface* (FaceUI, Touch) and *Number of events* (3, 4, and 5). Participants performed ten trials for each factor combination, resulting in 60 error-free timed trials per participant. The order of *Interface* levels was counter-balanced across participants. The order of *Number of events* levels was randomized within each *Interface* level. Participants were provided with practice trials for each combination until they felt comfortable operating the two interfaces. A study session lasted approximately 45 minutes.

### 6.3 Results

Our study task includes three sub-tasks: (i) selecting the requested date from the calendar, (ii) browsing event details for the selected date (i.e., switching between the detail views), and (iii) selecting the answer from the answer window. Therefore, we recorded the following: *start time* is the time the participant needed from tapping the start button to tapping the requested date from the calendar; *browsing time* is the time needed to browse and inspect the detail views, i.e., from selecting the correct date to tapping the check button to call in the answer window; and *selection time* is the time from tapping the check button to tapping the select button after having selected one of the options in the answer window. In addition, we recorded trials where participants submitted a wrong answer.

#### 6.3.1 Error trials, outliers, and total trial time

We observed that participants submitted the wrong answers in 41 trials (5.38%): 24 with the Touch interface (3.15%) and 17 with FaceUI (2.23%). A Wilcoxon Signed-Rank test showed no difference between the two *Interfaces*. To analyze the time results, we first removed all erroneous trials and then removed eight outlier trials with a total trial time (start time + browsing time + selection time) outside of ± 3 SD. Overall, the total trial time was 11.6% faster with FaceUI than with Touch (FaceUI 15.2s and Touch 17.2s). We used Repeated Measures ANOVAs and Bonferroni adjusted post-hoc pairwise comparisons to analyze the different times measurements.

#### 6.3.2 Start time

Figure 7a shows the start time for the two *Interfaces* in the three *Number of events* conditions. There was no significant difference in start time between Touch and FaceUI ($F_{1,11} = 0.04$, $p = 0.84$). The mean start time was 2.31s (SE 0.31) for Touch and 2.35s (SE 0.30) for FaceUI. There were also no significant differences in start time between the three *Number of events* conditions ($F_{2,22} = 0.26$,

$p = 0.78$). Conditions with 3, 4, and 5 events took 2.28s (SE 0.30), 2.34s (SE 0.28), and 2.38s (SE 0.31), respectively. There was also no significant interaction effect ($F_{2,22} = 0.61$, $p = 0.55$).

#### 6.3.3 Browsing time

Figure 7b shows the browsing time for the two *Interfaces* in the three *Number of events* conditions. The browsing time was significantly different between the two interfaces ($F_{1,11} = 6.52$, $p < 0.05$) and between the three levels of *Number of events* ($F_{2,22} = 39.55$, $p < 0.001$). FaceUI, with a mean Browsing time of 9.4s (SE 0.56), was significantly faster than touch interfaces, with a mean Browsing time of 11.4s (SE 0.77). Across the three *Number of events* conditions, FaceUI (9.4s, SE 0.56) was 17.5% faster than Touch (11.4s, SE 0.77). Bonferroni adjusted post-hoc pairwise comparisons between the three levels of *Number of events* showed significant differences (all *p's* < 0.001): unsurprisingly, with more events to inspect the longer time it took (3 events, 7.95s, SE 0.49; 4 events, 10.24s, SE 0.59; and 5 events, 13.09s, SE 0.82). The two factors did not interact ($F_{2,22} = 1.70$, $p = 0.21$).

#### 6.3.4 Selection time

Figure 7c shows the selection time for the two *Interfaces* in the three *Number of events* conditions. We observed no significant difference in selection time between Touch and FaceUI ($F_{1,11} = 0.92$, $p = 0.36$). The mean selection time was 1.59s (SE 0.06) for Touch and 1.64s (SE 0.04) for FaceUI. There were also no significant differences in selection time between the three levels of factor *Number of events* ($F_{2,22} = 1.28$, $p = 0.30$). Conditions with 3, 4, and 5 events took 1.61s (SE 0.06), 1.59s (SE 0.05), and 1.66s (SE 0.04), respectively. The two factors did not interact ($F_{2,22} = 0.80$, $p = 0.46$).

#### 6.3.5 Subjective feedback

Participants had prior experience with touch interfaces and were very comfortable using smartphones. One participant mentioned: *"I have been using touch interfaces on smartphones for a long time, I feel comfortable with them"*. We also observed a bias for the Touch interface when we asked participants to rate the two interfaces according to their overall preference on a 5-point scale. Touch was rated higher (mean rating 4.5, SD 0.5) than FaceUI (mean rating 3.5, SD 1.1). However, participants acknowledged that the concept of FaceUI was utterly new and interesting, and said they were not familiar with any similar concepts. One participant commented: *"It's a new method and I don't have experience with it. However, it seems a potential method for operating smartphones"*. Along these lines, we believe that once spatial interfaces with front-facing camera input, such as our FaceUI-calendar, become available on commercial smartphones people will feel comfortable using them.

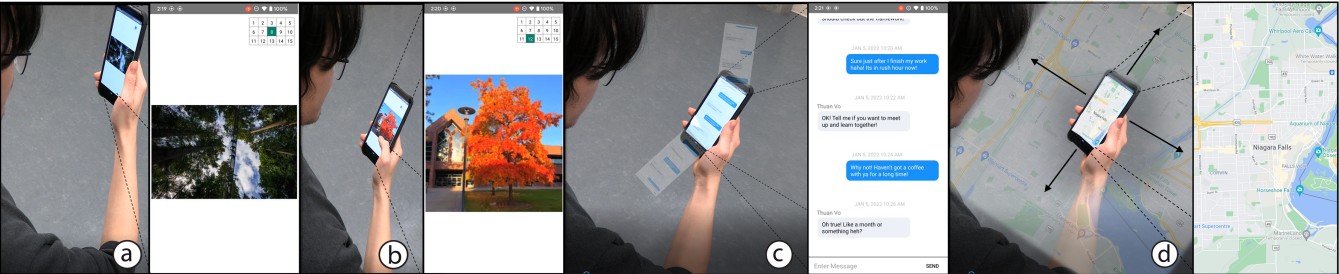

Figure 8: FaceUI enabled applications. A user: (a-b) browses through an image collection moving the phone sideways and up and down; (c) scrolls through the message history moving the phone vertically; and (d) navigates a map moving the phone across a stationary virtual map.

## 6.4 Summary

All our participants had more than nine years of prior experience using the traditional touch-based interfaces provided on smartphones, whereas FaceUI interactions were a new for them. Despite this, our results demonstrate that the FaceUI approach offers faster access to information than the touch-based alternative. Unsurprisingly, we found that both interfaces' start and selection times were comparable; the corresponding parts required touch input (i.e., tapping on date entries and buttons) in both interfaces. Instead, the difference between the two interfaces concerns the way they provide access to different calendar events' detail views and how the user can switch between these views. FaceUI enables the user to quickly retrieve spatially located virtual windows by moving the phone inside a large space. In contrast, touch interfaces require the user to switch between windows using frequent taps on small buttons or by swiping across the screen. Our participants could browse the event information significantly faster using the FaceUI approach than with the touch interface (Figure 7b).

Backed by our study results, we believe that FaceUI is a promising strategy that serves well as a supporting input technique in combination with conventional touch input. The study task required the participants to frequently alternate between touch input and spatial FaceUI-input during the study trials. Apparently, they could do so effortlessly.

## 7 DESIGN GUIDELINES AND FACEUI APPLICATIONS

We first summarize and present our key findings as design guidelines for FaceUI-enabled interfaces then we present three demo applications.

### 7.1 Design Guidelines

Our investigation offers the following guidelines to designers of FaceUI interactions and interfaces.

**Movement distance**. We found that participants preferred moving their hands within 40 cm from the face. Using this space for accessing mid-air regions will also help minimizing concerns related to arm fatigue. Thus, we recommend designers to limit the use of mid-air space to maximum 40 cm away from the user's face. Within this range, both users with short and users with long arms can comfortably access the space.

**Movement direction and mid-air regions**. Results indicate that participants preferred moving the phone in the horizontal direction over the vertical direction. Accordingly, designers should prioritize horizontal movements and try to avoid interactions that require the user to perform extensive movements up and down in front of the face. In addition, participants reported difficulties accessing areas far up "above" the nose, 20° above the nose or less works good.

**Viewing angle**. Designers do not only need to consider the ease with which users can physically move around and access areas in the mid-air space. Designers also need to consider how well the user can see the screen content when holding the phone in mid-air space. In general, positions further away from the mid-point in front of the user's face provide less comfortable viewing angles. Our study results suggest that viewing angles between -40° (left of the nose) and +40° (right of the nose) in the horizontal direction allow for clear and unconstrained visual access to screen content. In the vertical direction the range should be limited to +20° (above the nose) to -30° (below the nose). Extreme viewing angles warrant caution: they are only suitable for presenting screen content that can be grasped with a quick glimpse. Furthermore, extreme viewing angles can cause eyestrain.

**Mid-air space for browsing-intensive tasks**. Study 3 results showed that, compared to traditional touch-only interfaces, blending touch with mid-air interactions is an effective combination to quickly browse through and visually inspect different parts – or views – in an application. This is primarily due to the minimum switching costs involved with navigating between the parts, e.g., small device movements in front of the face for FaceUI vs. taxing taps on navigation buttons in a touch interface. Accordingly, we suggest that designers consider using the FaceUI approach to offload interactions into mid-air space particularly for lengthy browsing-intensive tasks that include frequent switching between different data views or windows. The more switching is required, the larger advantage can be expected from using the FaceUI approach.

### 7.2 FaceUI-Enabled Applications

We designed and implemented the following three applications to demonstrate how the FaceUI approach can be used on an off-the-shelf smartphone.

**Image Browser**. The FaceUI implementation in Study 3 did not consider scenarios where interface windows, views, or items are located in a grid in mid-air. However, many application would likely benefit from such as an item arrangement style. Consequently, we developed an image browsing app that offloads a set of images into a 3×5 grid in mid-air space in front of the user. The user can browse images by simply moving the device in the horizontal and vertical direction. While browsing the images, the user can tap the screen to access further details about an image. Figure 8a-b demonstrates the app scenario.

**Message History**. Scrolling through long lists of items with touch interfaces can be cumbersome and time-consuming, especially when using one-handed interaction mode [30]. We developed an application leveraging face-centered interactions to scroll through messages in a messenger app using one hand. In our implementation, the user can scroll through messages by moving the phone vertically between +20° up and -30° down (Figure 8c). Clutching works by pressing down on the screen and repositioning the device in mid-air space. When having released the press, the following device movements scroll the message list.

**Map Navigation**. We also designed and implemented a FaceUI-enabled map application where the user moves the phone in mid-air

to navigate across the map and to zoom the map. Phone movements going left, right, up, or down (relative to the user's face) are associated with movements in the corresponding directions *over* the map which remains stationary *under* the phone, as visualized in Figure 8d. The user can zoom in or zoom out by changing the distance between the face and the device. The user can also reposition the map under the screen. First the user grabs hold of the map by a pressing down on the screen and then moving the device, and so also the map. When the user releases the screen, the map is also released and anchored at its new position.

## 8 LIMITATION AND FUTURE WORK

We have only started our exploration of the FaceUI approach and our current work has limitations. Accordingly, we see a need for further investigations and also plenty opportunities for interesting future work. For a start, FaceUI requires users to move the phone in mid-air to access screen contents, which may cause eye and arm fatigues for prolonged use. Further investigation is needed to explore solutions to minimize such fatigues. Suggestions and design guidelines for reducing arm fatigue provided by prior research [22] should be examined also for FaceUI scenarios. Related to this, we asked our participants to keep the head still and to only move the phone in the studies but further studies could explore possible ways to leverage the combination of head and smartphone movements to access in-air regions in a way that does not causes any arm fatigue.

Using FaceUI in public spaces may trigger feelings such as embarrassment or discomfort due to hand movements which may attract by-passers' undesired attention. Thus, it is also important to carry out studies to explore the social acceptance of performing FaceUI-enabled interactions in public (and private) spaces. Moreover, when using FaceUI, on-screen information can be viewed by surrounding people, which may triggering privacy concerns. Future studies need to investigate users' and by-passers' privacy concerns. Future studies could also explore strategies to implement software and hardware-based privacy filters that keep screen content only visible to the user.

We only investigated the performance of FaceUI when accessing application windows located in the horizontal direction. Further investigations are needed to explore natural delimiters to switch between FaceUI and touch and the performance of FaceUI-enabled applications, where windows are arranged in both horizontal and vertical directions, such as in our image browser (Section 7.2).

## 9 CONCLUSION

We have presented FaceUI, a novel interaction approach for smartphones which leverages mid-air space to access face-centered spatial user interfaces. The FaceUI approach is basically a self-contained position tracking mechanism for off-the-shelf smartphones that uses the smartphone's front-facing camera to track the phone's position relative to the user's face. Through two user studies, we first explored different factors that influence the design and performance of applications relying on FaceUI. Based on the results, we designed a FaceUI calendar app. We then compared users' performance using this calendar with a touch-based calendar interface in analytical tasks where users searched for and compared different calendar events. Results showed that FaceUI is a promising approach that enables fast access to parts of interfaces by off-loading these parts into mid-air space in front of the user where the user can easily switch between the parts by moving the phone to the corresponding in-air location, which displays the part on the screen.

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
