# OpenReview forum: "FaceUI: Leveraging Front-Facing Camera Input to Access Mid-Air Spatial Interfaces on Smartphones "
_graphicsinterface.org/Graphics_Interface/2022/Conference — GI 2022_

### Official Review · Reviewer_HZYa · 2022-04-12
**Interesting concept with some statistical issues in studies**

**Rating:** 6
**Confidence:** 3

**Review:**

This paper presents FaceUI, an interaction concept that uses a smartphone’s facial tracking features to enable spatial interactions. FaceUI allows for users to interact in 3D space by simply moving the phone relative to the user’s head. This technique is validated in three studies: the first establishes the comfortable and usable horizontal, vertical, and distance bounds for optimal FaceUI usage; the second measures the fatigue and readability ramifications of various angles of the phone relative to the face, and the third measures usability of FaceUI in a more realistic task.

Overall, I find the concept of FaceUI interesting and novel. Smartphones have had facial sensors and face tracking for a long time at this point, and barring unlocking (e.g. FaceID) and aesthetic Snapchat filters, not much has been done in making the face and relative smartphone position an actual meaningful artifact in the system. This paper does a good initial step in establishing the realistic and applied bounds that such systems could use to create more usable smartphone systems.

However, I have a few issues with how the user studies were conducted and analyzed. I will list them here:

- Section 4.3.3: I would have liked to see an analysis of error rate between the different distances, and not just aggregated overall.
- While it was addressed in the limitations, it should be made clear from the beginning that users were asked to keep their head still for the studies. I didn’t get that at first.
- I would have liked to see a better or more established scale for qualitative answers, rather than just a 1-5 Likert scale. See scales such as the NASA-TLX or System Usability Scale.
- Also on the topic of qualitative questions, I would have liked to see standard deviations for these likert answers.
The studies need to be counterbalanced to avoid any bias in a within-subjects study. Participants doing Up then Left could see advantage over Down then Left (for example), and counterbalancing with a Latin square can smooth these differences out.
- Section 5.5: “The results show that Up is slower than when the phone is positioned in any other region” - watch this wording. The Study 2 results showed that Up was significantly slower than Down, but from what I understood, there were no other significant differences otherwise, so generalizing like that is not accurate.
- On that note, (*) indicators on the graphs to show which pairs or values were statistically significant would be helpful.
- Section 6.3.2 - if there was no statistical significance I don’t see the value in saying something was slightly faster, as the effects may just be random noise.
- Typos: 6.3.3 - upside down ! instead of > or <, 6.3.5 “one participants”

Overall I find this concept interesting and the studies generally convincing barring a few statistical issues I would want addressed. This paper is marginally above my acceptance threshold but these issues I outlined are important.

---

### Official Review · Reviewer_qh7T · 2022-04-13
**Interesting work on enabling mid-air interactions with smartphone camera tracking**

**Rating:** 8
**Confidence:** 3

**Review:**

This paper presents a mid-air spatial interface that uses smartphone front-facing camera to track the phone position regarding the user's face. In this way, users can browse information by moving the phone in the space in front of their faces. The authors claimed that no previous work explored such face-centered in-air interactions with smartphones. The authors run users studies to explore different design parameters as well as to compare the face-based mid-air interactions with touch-based interactions.

Overall, the paper makes important contributions as seems to introduce a novel interaction technique. Extensive studies are done and results are analyzed sufficiently. I also really like the discussions that summarize the design guidelines and possible applications.

The paper could benefit from discussing more qualitative feedback from participants including the limitations of these interaction techniques and the challenges participant faced. It may not be always practical to move the smartwatch around with good precision or people may feel musle fatigue through the repetitive movement of hands. More discussions on those aspects were desired.

---

### Official Review · Reviewer_eZU7 · 2022-04-14
**Good paper, misleading title**

**Rating:** 7
**Confidence:** 4

**Review:**

The writing is clear and the structure is good. The contributions are:

1. the clever idea of using the selfie-camera to estimate radial Z, giving a cylindrical coordinate system using accelerometers for XY
2. the trivial implementation of #1
3. the exploration of the design space and some demos
4. quantitative analysis of the UI

As with most UI/HCI papers, the strength of the contribution and the science is #3 and #4, not the underlying idea and implementation.

I have only casual familiarity with the area of camera-driven interfaces. I found the related work description to be good and appropriate based on that knowledge.

The analysis has a moderate subject size (12) and little control for confounding factors/lack of diversity in the subject pool. I think that is acceptable for the GI venue and nature of the interface. For followup work or with a larger contribution I would suggest ensuring a larger sample size and specifically considering issues such as lighting, skin tone, facial features, age (affecting head size!), culture (affecting comfort distance for holding the phone), etc.

The methodology is clear and the results are sufficient to convince me that "this works and should be published", although not enough to decide whether I would choose this mechanism over other input schemes for a given application.

The "face" aspect of this paper is that it uses the front-facing camera to estimate distance along the Z-axis for 3D positioning of the phone relative to an augmented realityUI. That is a contribution of the paper, and it is a clever one, but "FaceUI" is misleading as a title because it makes it seem like the UI is the face (eg., smile to press OK) or that it will be a UI for a face (e.g., avatar facial mimicking). I would simply delete FaceUI from the title and elsewhere in the paper because the long-form explanation in the title is clear.

---

### Decision · Program_Chairs · 2022-04-17

Accept